# Climate Action at Public Health Schools in the European Region

**DOI:** 10.3390/ijerph18041518

**Published:** 2021-02-05

**Authors:** Rana Orhan, John Middleton, Thomas Krafft, Katarzyna Czabanowska

**Affiliations:** 1Department of International Health, Care and Public Health Research Institute CAPHRI, Faculty of Health, Medicine and Life Sciences, Maastricht University, Duboisdomain 30, 6229 GT Maastricht, The Netherlands; kasia.czabanowska@maastrichtuniversity.nl; 2School of Health and Related Research, University of Sheffield, 30 Regent Street, Sheffield S1 4DA, UK; 3Association of Schools of Public Health in the European Region (ASPHER), Avenue de Tervueren 153, 1150 Brussels, Belgium; johnmiddleton@phonecoop.coop; 4Department of Health, Ethics and Society, Care and Public Health Research Institute CAPHRI, Faculty of Health, Medicine and Life Sciences, Maastricht University, Universiteitssingel 40, 6229 ER Maastricht, The Netherlands; thomas.krafft@maastrichtuniversity.nl; 5Department of Health Policy Management, Institute of Public Health, Faculty of Health Sciences, Jagiellonian University, 31-066 Krakow, Poland

**Keywords:** climate change, climate action, public health education, university social responsibility

## Abstract

Climate change is putting the achievement of all Sustainable Development Goals at risk and leads to negative impacts on human health and well-being. Consequently, tremendous social responsibility lies with public health professionals and their associations. Therefore, this study addressed the following question: “How can the Association of Schools of Public Health in the European Region (ASPHER) best support the goals of the European Green Deal through its network of public health schools and departments?” This study looked at the implementation of climate education in public health schools in the European region and climate action taken by these public health schools. An online survey among ASPHER members with a 51% overall response rate (excluding non-European members) shows that 64% of the responding schools provide climate-health educational offerings, while 63% consider these for the future. Additionally, most climate actions taken by the schools were ad hoc actions. These findings show that a systematic approach is missing, and there is a general lack of strategy in most schools. We consequently recommend that schools invest in climate and health education in their curricula and become exemplars for climate action to actively contribute to the achievement of Europe’s climate goals.

## 1. Introduction

Human-induced climate change is putting the achievement of all Sustainable Development Goals (SDGs) at risk [1]. Any increase in global warming leads to negative impacts on human health and well-being: directly affecting the social and environmental determinants of health and indirectly through consequences of climate change such as migration, conflicts, and political instability [1]. Under an optimistic socio-economic scenario, the World Health Organization (WHO) expects over 250,000 additional deaths per year globally between 2030 and 2050 [2]. Due to the complexity of the causal pathways, however, the actual number of affected people will most likely be hundreds of millions [2]. Particularly vulnerable groups are at risk, and social vulnerability and exposure to adverse health impacts vary across Europe, widening health inequities in the region [3].

In 2016, European Union Member States were responsible for 7.34% of global greenhouse gas emissions, which are the driver of human-induced climate change [4]. At the United Nations Framework Convention on Climate Change Conference of the Parties (UNFCC COP) in December 2015, under the so-called ‘Paris Agreement’, the Union committed to reducing greenhouse gas emissions by at least 40% by 2030 compared to 1990 [5]. This reduction would prevent 74,000 premature deaths across the European region in 2030 [6]. By setting out the European Green Deal in December 2019, the European Commission aims for a reduction of 50% by 2030 [7]. The Climate Law will regulate this goal and the Union’s commitment to a climate-neutral Europe by 2050 [7]. Greenhouse gas emissions were reduced by 23% in the last two decades [4]. However, to reach the 2030 and 2050 goals, an upsurge of climate actions is needed across all sectors, both local and global [1,8]. One of the areas for improvement is the reinforcement of the links between public health and climate adaptation [3].

In the ‘State and Outlook 2020′ report, the European Environment Agency states that the link between climate change and health is complex [3]: The gaps and uncertainties in evidence make it arguable whether negative health impacts can be reduced by current policies [1]. An improvement of the collective understanding of climate and health, and more evidence-based and efficient communication regarding the health risks, efficient solutions and strategies, their costs, and effective implementation are needed to develop and implement proper mitigation and adaptation policies [1,9]. Due to the complexity of climate change, it is essential that future generations of public health professionals have the knowledge to be able to work on and address the uncertainties faced currently. With the view on Health in All Policies, public health professionals need to engage with various stakeholders in the debate on the intricate pathways of direct and indirect adverse effects of climate change on health. Therefore, climate and health education has to be included in the public health curriculum.

A study by Krasna et al. shows that the labour market for public health graduates with training in climate change is emerging [10]. The growing need for graduates with training in climate change and health requires public health schools to include climate change and health in their curricula. At the same time, universities are change leaders and have a social responsibility role in helping society respond to the threat to public health [10,11]. According to the WHO’s global strategy on health, environment and climate change, universities are one of the crucial settings for interventions [12]. The role of educational institutions is (1) to ensure a safe environment for climate education; (2) to generate awareness about the link between environment and health; (3) to provide education on sustainable approaches; and (4) to be the facilitator in the inclusion of best practices in the wider community. By including climate change and health competencies in public health curricula, universities can contribute to achieving the Union’s 2030 and 2050 goals. Public health graduates with training in climate change and health can influence educational institutions from within [10] and reduce the risks and impacts of climate change for European citizens’ health and well-being, by bringing skills and knowledge to the public health workforce.

Therefore, in the past years, Civil Society Organisations (CSOs) initiatives to include climate change and health in public health curricula have arisen. The Global Consortium on Climate Change and Health (GCCHE) proposed “Core Climate & Health Competencies for Health Professionals” in 2018 [13]. In the same year, the Association of Schools of Public Health in the European Region (ASPHER) listed three competencies related to climate change in the subsection on “Population Health and Its Material—Physical, Radiological, Chemical and Biological—Environmental Determinants” in the 5th edition of European List of Core Competences for the Public Health Professional (2018) [14]. In 2016, the Council of Academic Public Health Institutions Australia (CAPHIA) listed two competencies related to climate change in their Foundation Competencies for Public Health Graduates in Australia from 2016 [15]. The most recent addition to CSO-led initiatives is the WHO-ASPHER Competency Framework for the Public Health Workforce in the European Region, which states one competency on climate change: “Knows and correctly identifies the main features of the climate change process, along with its implications for public health, and understands the public health responsibility for the natural environment” [16].

With the European Green Deal, the Commission commits to preparing a European competence framework for schools, training institutions, and universities to be able to develop and assess attitudes, skills, and knowledge on climate change and sustainable development [7]. The Ostrava Declaration of the WHO European Region specifies that health aspects of climate change should be included in education curricula, non-formal education, and workforce continuing professional education [17].

Although climate and health education has been a topic of interest in the past years, it is unknown to which extent this has been reflected in the continuous updates of European public health schools’ curricula. To our knowledge, there are also no studies conducted assessing public health schools’ climate action. ASPHER is the European organisation dedicated to strengthening the role of public health by improving education, building capacity in public health, and raising awareness about new developments that need to be addressed by educational interventions such as climate action. This study focused on ASPHER and its members, which are public health schools in the European Region and associated partners in other parts of the world. This study addresses the following question: “How can ASPHER best support the goals of the European Green Deal through its network of public health schools and departments?” This pilot study particularly looks at (1) the implementation of climate education in public health schools in the European region, and (2) climate action taken by these public health schools. Climate action is defined as “stepped-up efforts to reduce greenhouse gas emissions and strengthen resilience and adaptive capacity to climate-induced impacts” [18]. The results of this study will be valuable to public health schools, both in and outside the European region, and other (health-related) schools in the implementation of climate change and health education in their curricula and taking climate action themselves. Furthermore, the results will contribute to ASPHER’s future strategies and may prove useful in developing the European competence framework by the European Commission.

## 2. Materials and Methods

Due to this study’s two-fold character, the online survey was split into two parts addressing: (1) climate action, and (2) climate education. The first part was developed by adapting parts of the Auditing Instrument for Sustainability in Higher Education (AISHE), an assessment tool recognised by the Dutch and Flemish Accreditation Organisation (NVAO) for sustainable development in higher education institutions [19]. This assessment tool consists of statements that correspond with a level of climate action: ad hoc (Level 1), cohesive (Level 2), systematic (Level 3), collaborative (Level 4), and an example for others (Level 5). The statements and the corresponding climate action levels are listed in Appendix B. The second part of the survey, on climate and health education, was designed by the Global Consortium on Climate and Health Education (GCCHE) Coordinating Committee for a survey on the state of climate and health education among their 160 institutional members, including international health professions schools and programs [20]. The current study has been ethically approved via the University of Sheffield’s Ethics Review Procedure, as administered by the School of Health and Related Research (ScHARR; Application 034030).

This study has been conducted among members of the Association of Schools of Public Health in the European Region (ASPHER). ASPHER currently has 89 full members and eight associate members. These members are schools, research institutes, and other structures with a role in education and/or training in public health, active in the European region as defined by the World Health Organization (WHO) [21]. This study had a particular focus on schools, but all full members were approached to fill in the survey. Current associate members are not based in the European region and, therefore, were excluded from participation. The primary contact person from the ASPHER member served as the survey respondent. In case they could not fill in the survey, other representatives from the member were accepted as the respondent. The online survey was open for members from Monday 4 May until Monday 8 June 2020, and it was sent to members via the newsletter and via general and personalised reminders to the members mailing list. These emails and the survey were accompanied by a participant information sheet, where voluntary participation and data protection were covered. The online survey also included a consent form.

The survey consisted of 40 questions (see Appendix A for the survey questions). Four questions were about the participant’s details to confirm which ASPHER member they represent; 15 questions about the member’s climate action; 20 questions about the inclusion of climate education in their curricula; and one question about any questions and/or remarks.

## 3. Results

With 45 out of 89 ASPHER members, the overall response rate was 51%. The responses represent public health schools from 24 out of 53 countries in the WHO European region.

### 3.1. Climate Education

#### 3.1.1. Climate-Health Educational Offerings

Members were asked about their climate and health educational offerings, assessment of climate and health knowledge, and whether they have received evaluations from students on their experience of and/or satisfaction with the climate-health teachings. Their responses are listed in Table 1. The schools of 29 of 45 respondents (64%) offer climate and health education, and 16 schools (36%) do not. Out of 28 schools that offer climate and health education, eight schools (29%) offer planetary health modules, courses or programs: In five of these schools, the planetary health and climate-health modules or programs are linked or integrated; in three schools, they are not. Thirteen of 26 schools (50%) assess their students’ climate-health knowledge. The most common assessment methods are exams (9 out of 13 [69%] respondents), papers (54%), and quizzes (38%).

Out of the 29 schools that offer climate and health education, 21 schools (72%) provide a climate-health session as part of a compulsory core course, and six schools (21%) do so as part of a non-compulsory course. Five schools (17%) have a climate-health stand-alone elective course, while two schools (7%) additionally provide this as a required course. None of the schools offer a climate-health master’s or certificate program, a climate-health doctoral program, or climate-health post-doctoral positions.

Out of the six schools with a standalone course on climate and health, four schools provide online tutorials or Massive Open Online Courses (MOOCs), three schools provide in-class exercises, and three schools provide lectures on climate and health. The course credits range from 1.6 to 10 European Credit Transfer and Accumulation System (ECTS) credits. The climate-health educational offerings have been in place for an average of 8 years, ranging from less than 1 year to more than 30 years. Seventeen of 27 respondents (63%) stated that their school has received evaluations from students on their experience of and/or satisfaction with the climate-health teachings.

Twenty-four respondents described the main goals of their climate and health curriculum, which include:To develop an understanding of the concepts of “climate change” and “sustainability”, the process of climate change, its consequences on populations, and its health impactsTo develop an understanding of the public health challenges of climate changeTo develop an understanding of the challenges and to identify (multisectoral) approaches and solutionsTo be equipped with the skills to do a comprehensive critical analysis and policy formationTo increase the visibility of the United Nations 2030 Agenda and Sustainable Development Goals (SDGs)To get experience in climate and health leadershipTo relate climate change mitigation to healthcare and the role of healthcare professionalsTo raise awareness and spark interest in climate changeTo stimulate action to reduce the carbon footprint

#### 3.1.2. Developing Climate and Health Education

Members were asked about any climate and health offerings under discussion to add, positive responses to adding them, what they found helpful in instituting or developing climate-health curriculum, any challenges they faced, and whether the school has partnerships. Their responses are listed in Table 2.

Twenty-four of 38 respondents (63%) stated that climate-health offerings are under discussion to add. Seventeen of the respondents (45%) consider a session as part of a required core course, and eight schools (21%) do as part of a non-required course. A climate and health standalone elective course is under discussion by five schools (13%), as it is for a standalone required course.

Positive responses to adding climate and health to the curriculum were from students (26 of 35 [74%] respondents), faculty (46%), and administration (17%). Five schools (14%) stated not to have received a positive response.

Out of 35 schools, interest from students (83%) and interest from faculty (66%) helped institute or develop a climate and health curriculum. Interest from administration (37%), support from board members (37%), and support from donors (14%) were also helpful factors.

Among the most common challenges in trying to institute a climate and health curriculum was lack of available staff to work on its development (17 of 38 [45%] respondents), lack of funding/time to support its development (29%), no available space in the core curriculum (24%), lack of teaching materials and staff expertise (21%), and competing institutional priorities/politics (16%). Lack of interest or demand from students (8%) and administration or other scepticism about climate and health science (8%) were least common. Thirteen of 38 schools (34%) stated not to have encountered any challenges.

Out of 26 respondents that offer climate and health education, 15 (58%) have a partnership on climate change and human health; eight schools (31%) have a partnership with a non-academic institution; 10 (38%) with another academic institution on research; and 4 (15%) with another academic institution on training.

### 3.2. Climate Action

In 10 aspects of climate action, members were asked to choose the most applicable statement to their school. Their responses are listed in Table 3.

#### 3.2.1. School’s Vision

Out of the 27 respondents that state that climate action is included in the school’s vision (60%), 15 (56%) state that this is implicitly applied, e.g., via the university’s vision (Level 1: Ad hoc); two (7%) have an explicit vision on climate education, which is being applied within the majority of the programme (Level 2: Cohesive); six (22%) have a vision on climate education, which is visible in the profiling of the school or the educational programme and keeps this vision updated (Level 3: Systematic); two (7%) actively collaborate with the professional field and centres of expertise in developing their vision on climate action and its periodic updating (Level 4: Collaborative); and one (4%) is a recognised pioneer in translating the concept of climate action to the educational domain (Level 5: An example for others).

#### 3.2.2. Strategy

Out of the 24 schools that included climate action in their strategy (53%), seven (29%) state that this is implicitly applied, e.g., via the university’s strategic plans, and seven (29%) state that different actors within the school formulate a strategy and objectives for their activities (Level 1: Ad hoc). Eight out of 24 (33%) have an explicit view on climate action translated to concrete objectives within several policy areas (Level 2: Cohesive); one (4%) periodically evaluates and reflects the realisation of the concrete objectives and states that the evaluation takes place in coherence with the vision on climate action (Level 3: Systematic); one (4%) actively collaborates with the professional field and centres of expertise in developing its strategies on climate action and its period updating (Level 4: Collaborative); and zero schools (0%) are a recognised pioneer in translating the concept of climate action to several policy areas (Level 5: An example for others).

#### 3.2.3. Personnel

Thirty-one out of 45 (69%) state that a few employees have knowledge in the field of climate action in the educational domain and that the school provides the opportunities to execute individual educational initiatives (Level 1: Ad hoc). Six out of 45 (13%) state that expertise in the field of climate action is broadly present in the school and that this expertise is being kept updated systematically (Level 3: Systematic).

#### 3.2.4. Networks

Out of 44 respondents, 26 (59%) state that contact with the professional field and/or centres of expertise in the field of climate action are limited to individual employees (Level 1: Ad hoc), while eight schools (18%) state that this contact is at the school level (Level 2: Cohesive). In three schools (7%), the school develops its network based on the desired exchange of knowledge and expertise about climate action and the training hereof (Level 3: Systematic).

#### 3.2.5. Culture

Thirteen respondents (30%) state that the school takes measures on a number of points to achieve common values for climate action (Level 1: Ad hoc). There is a coherent policy in 10 schools (23%) aimed at developing these common values (Level 2: Cohesive). In four schools (9%), the school is recognised by a culture in which values linked to climate action are central (Level 3: Systematic). One school (2%) develops common values for climate action with its partners (Level 4: Collaborative). Thirteen schools (30%) stated not to take any measures to achieve common values.

#### 3.2.6. Reducing the Ecological Footprint

Seventeen respondents (38%) state that the school takes measures on several points (Level 1: Ad hoc); 11 schools (24%) state to have a concrete policy aimed at reducing the ecological footprint of the school and at contributing to environmental restoration (Level 2: Cohesive); and eight (18%) state to do this systematically (Level 3: Systematic). Five schools (11%) stated that they take no measures with regard to the physical environment, however, another question specified at the reduction of the ecological footprint shows that this number is two schools (4%). The measures taken by schools to reduce their ecological footprint are listed in Table 4. The most common measures are improving waste management (34 of 45 [76%] responses), raising awareness among students (not by formal education) (62%), raising awareness among personnel (58%), avoiding waste (56%), increasing re-use, repair or recycling (56%), providing personnel (50%) and students (38%) with the opportunity to be involved in sustainability, and a green travel policy (36%). Five out of 45 schools (11%) have a net-zero carbon building. Other responses (2%) included “modernizing the building infrastructure and improving [energy] efficiency making concrete investments, including green energy sources”. Four schools (9%) stated that their school divested from fossil fuel companies or that their school is fossil fuel free. However, when explicitly asked whether their school accepts donations and grants from fossil fuel companies or other environmentally destructive companies, 26 out of 44 respondents (59%) said “no”. One school (2%) said “yes”, and 39% (17 respondents) do not know.

#### 3.2.7. Communication

Twenty-six respondents (58%) state that the school’s communication about climate action is incidental and involves separate activities. Internal and external communication run parallel to the activities carried out on climate action in eight schools (18%). In four schools (9%), the school has and implements an explicit communication policy on climate action.

#### 3.2.8. Education and Research

Out of 45 respondents, 33 (73%) state that the perspective of climate change in research and practical assignments depends on the individual teacher or students (Level 1: Ad hoc). For research projects, this is the case in 84% (38 respondents). In four schools (9%), each student carries out a practical assignment at least once from the perspective of climate change, while for studies that are being conducted, an integral perspective by each student is ensured in one school (2%) (Level 2: Cohesive). This integral perspective is being used in all assignments in the professional field in three schools (7%) (Level 3: Systematic). One school (2%) has this systematic approach for answering their research questions. See Section 3.1 for the implementation of climate and health education.

#### 3.2.9. Innovation

Twenty respondents (45%) state that the school’s research and/or graduation projects occasionally lead to innovative solutions for issues associated with climate change (Level 1: Ad hoc). In 12 schools (27%), the school stimulates such projects (Level 2: Cohesive), while this is being done systematically in 0% of the schools (Level 3: Systematic). Seven schools (16%) do not have research and/or graduation projects on climate change that lead to innovative solutions.

## 4. Discussion

Through our survey among 45 member schools, we have obtained a view on the implementation of climate education and climate action by public health schools in the European region. The assessment of 10 aspects of climate action including education shows that most of the 45 surveyed public health schools are taking action to address the issue of human-induced climate change. At the same time, our results show that most of these actions are at the level of ad hoc actions. A systematic, collaborative approach is largely missing, and there is a general lack of strategy in most schools.

Climate action starts by recognising the urgency of the problem and the responsibility that the public health field including schools carries [8]. This recognition should be solidified in the school’s vision and strategy and through common values for climate action. However, we found that most public health schools apply climate action only implicitly in their vision and strategy and do not have common values for climate action or do not show any effort in achieving them. Some schools have a visible vision in the profiling of the school and have this view translated to concrete objectives within several policy areas. We encourage these cohesive and systematic actions at other schools to take a step towards what Krasna et al. argue: “the issues of sustainability are so far-reaching that it can be argued that educational institutions must reframe their full mission, using sustainability as their foundation” [10].

At the same time, public health schools have a key function in leading change in society and facilitating the inclusion of best practices in the wider community [9,10,12]. This role can be played at the level of communication, physical environment, and innovation. As for communication, our findings show that public health schools mostly communicate incidentally about climate action. This is in contrast with the need for more evidence-based and efficient communication for developing and implementing proper mitigation and adaptation policies [1].

As for the physical environment, most schools take measures to reduce their ecological footprint by improving waste management, implementing a green travel policy, and providing students and personnel with the opportunity to be involved in sustainability. Although most measures were not strategic actions, some schools were taking measures together with their partners, and some schools have innovative solutions for reducing the ecological footprint and are an example for others. Also, most schools raise awareness among their students and personnel, which aligns with the role of educational institutions, according to the WHO [12]. Since European Union Member States were responsible for 7.34% of global greenhouse gas emissions [4], these are steps that we encourage other schools to implement. Notable is that we found that five schools have a net-zero carbon building, which is a needed practice: buildings are responsible for 40% of the consumed energy and 36% of carbon emission in the European Union [22]. Reducing the ecological footprint is a needed practice for meeting and strengthening the commitments under the Paris Agreement and reducing the direct and indirect detrimental impacts of climate change on human health and well-being [1]. We regard commitment to climate action in the physical environment as a key didactical concept to upsurge social responsibility in the public health field.

As for innovation, our results show that schools are not sufficiently stimulating projects that can lead to innovative solutions for issues associated with climate change. We argue that this is a key aspect in leading change in society, and therefore this needs more attention from schools.

Our results also show that climate and health education is currently still falling short of the actual needs and in many cases depending on individual teachers and students: 29 out of 45 schools have provided climate and health education since an average period of 8 years, but most of these schools offer a climate-health session as an integrated component of a course rather than a stand-alone course. We argue that these deficiencies insufficiently respond to the growing demand for public health graduates with training in climate change and health [10]. In a recent narrative synthesis, Lee et al. stated that the accuracy of reported knowledge about the impacts of climate change varies according to the method employed and that there are erroneous ideas and misconceptions about causes of climate change [23]. This results in the fragmentation and lack of integration of the knowledge on different aspects of climate issues, which greatly affects its usability in policymaking process [24].” We acknowledge that not every public health professional needs to become an expert in climatology and ecology, but we argue that all graduates need to understand the basics, be able to communicate knowledgeably, and form partnerships and alliances in multidisciplinary settings with experts in these areas [25]. Therefore, interprofessional, interdisciplinary, and transformational educational activities, which require not only problem-solving skills but also system thinking, change and implementation strategies should be included in modern curricula addressing climate change. These competencies are recommended in the frameworks and competencies lists as proposed by the GCCHE, ASPHER, CAPHIA, and WHO-ASPHER Framework for the Public Health Workforce in the European Region [10,13,14,15,16], and Krasna et al. echo that the skills that are required in job postings are in alignment with these competencies.

However, our findings resonate with the future need, as most of the schools surveyed consider (additional) climate and health educational content and/or courses; specifically, stand-alone compulsory courses and new master programmes dedicated to climate and health are considered by some schools. We found that doctoral programs and postdoctoral positions are neither offered nor considered by public health schools. At the same time, Krasna et al. show that 21.4% of job listings in the field of climate change and health are offered by university/academia [10]. The lack of offerings of these programs and positions might therefore need reconsideration by schools.

Although the majority of the schools consider climate and health education, our results show that lack of staff, funding, and time needed for developing curricula are the biggest challenges in realising it. There is also a lack of didactical materials and experienced staff for developing such educational programmes.

At the same time, students and faculty proved to be helpful in instituting or developing climate and health curricula in schools that offer climate and health education. This stresses the importance of the European competence framework plan as stated in the European Commission’s Green Deal, which aims “to help develop and assess knowledge, skills, and attitudes on climate change and sustainable development” among others by providing support materials and by facilitating the exchange of good practices [7]. These challenges also put an emphasis on the facilitating and leading role that network organisations such as ASPHER and GCCHE have. To this end, ASPHER recently adopted its value “corporate citizenship—leading by example through ASPHER’s role and responsibilities as a steward for community and planetary health in partnership with a wide variety of actors working locally and globally to foster social responsibility” [26].

### Limitations

This study has several limitations. First of all, the response rate is 51%. This is most likely due to the short input period of 5 weeks and the COVID-19 pandemic that has demanded a lot from public health schools in the European region. Also, the fact that most questions required an answer led to fewer respondents, as some school representatives had filled in the survey up to 80%. The second limitation is that there is a potential for nonresponse bias, as was identified in the survey by Shea et al. from the GCCHE [20]: It is more likely that schools that already offer climate and health education, and additionally in this study, that already take climate action, fill in the survey. However, Hendra and Hill claim that there is little relationship between response rate and nonresponse bias. Additionally, there is no evidence to support the 80% response rate (or any other rate) as optimal [27]. To limit the nonresponse bias, personalised reminders were sent to non-respondents without any prior knowledge about their current activities or plans. As this was a pilot study, a repeated study would bring more insightful results.

## 5. Conclusions

Climate change is the biggest threat to public health. Any increase in global warming leads to negative impacts on human health and well-being. At the same time, the gaps and uncertainties in existing evidence make it arguable whether current climate adaptation and mitigation policies are effective in reducing the negative health impacts. The assessment of climate action including education shows that most of the 45 surveyed public health schools are taking action to address the issue of human-induced climate change. Yet, most of these actions are ad hoc: a systematic, collaborative approach is missing, and there is a general lack of strategy in most schools.

This study stresses the need for long-term investment again, way ahead of the 2030 target of the European Union. If the European target is to be delivered, all constituent elements of European life will need to play their part [8]. Climate and health education has to be included in the public health curriculum to improve the collective understanding of climate and health and allow proper mitigation and adaptation policies to be developed and implemented. At the same time, universities have a key function in leading change in society and, therefore, have a social responsibility. Public health schools need to review their strategies and information for students and personnel. We recommend that they look more formally at their potential contribution to the Green Deal target of a 50% reduction in carbon footprint by 2030 and a climate-neutral Europe by 2050. By an upsurge of their climate actions, schools can be active contributors to the achievement of the region’s climate goals.

## Figures and Tables

**Table 1 ijerph-18-01518-t001:** Climate-health educational offerings, assessments, and evaluations.

Survey Question	Response, No. (%)	Response, No. (%)	
		Yes	No
Does your school offer climate-health education?	45	29 (64)	16 (36)
What climate-health education does your school offer? ^1^	29		
Climate-health session as part of non-required course	6 (21)		
Climate-health session as part of required core course	21 (72)		
Climate-health standalone elective course	5 (17)		
Climate-health standalone required course	2 (7)		
Climate-health masters or certificate program	0 (0)		
Climate-health doctoral program	0 (0)		
Climate-health post-doctoral positions	0 (0)		
If your school offers a standalone course on climate-health, what teaching methods are used? ^1^	6		
Labs	2 (33)		
Lectures	3 (50)		
In-class exercises	3 (50)		
Online tutorials or MOOCs ^2^	4 (67)		
Internships outside the classroom	0 (0)		
Does your school offer planetary health modules, courses, or programs?	28	8 (29)	20 (71)
Does your school assess students’ climate-health knowledge?	26	13 (50)	13 (50)
How is climate-health knowledge assessed? ^1^	13		
Quizzes	5 (38)		
Exams	9 (69)		
Papers	7 (54)		
Capstone	0 (0)		
Thesis	2 (15)		
Dissertation	1 (8)		
Has your school received evaluations from students on their experience of and/or satisfaction with the climate-health teachings?	28	17 (61)	11 (39)

^1^ Respondents were asked to select all responses that apply. ^2^ MOOCs, Massive Open Online Courses.

**Table 2 ijerph-18-01518-t002:** Challenges, responses, and future plans for climate-health education.

Survey Question	Response, No. (%)
Are any climate-health offerings under discussion to add? ^1^	38
Session as part of non-required course	8 (21)
Session as part of required core course	17 (45)
Climate-health standalone elective course	5 (13)
Climate-health standalone required course	5 (13)
Climate-health masters or certificate program	2 (5)
Climate-health doctoral degrees	0 (0)
Climate-health post-doctoral positions	0 (0)
Nothing being considered	14 (37)
Have you received a positive response to adding climate-health curriculum? ^1^	35
Yes, from students	26 (74)
Yes, from faculty	16 (46)
Yes, from administration	6 (17)
No, have not received a positive response	5 (14)
Other	0 (0)
What have you found helpful in instituting or developing climate-health curriculum? ^1^	35
Interest from students	29 (83)
Interest from faculty	23 (66)
Interest from administration	13 (37)
Support from Board members	13 (37)
Support from donor	5 (14)
Other	3 (9)
Have you encountered any challenges in trying to institute climate-health curriculum? ^1^	38
Yes, lack of interest or demand from students	3 (8)
Yes, administration or other scepticism about climate-health science	3 (8)
Yes, lack of funding/time to support its development	11 (29)
Yes, lack of available staff time to work on its development	17 (45)
Yes, no available space in the core curriculum	9 (24)
Yes, lack of teaching materials and staff expertise	8 (21)
Yes, competing institutional priorities/politics	6 (16)
No challenges	13 (34)
Other	4 (11)
Does your school currently have any partnerships on climate change and human health?	26
Yes, with another academic institution on training	4 (15)
Yes, with another academic institution on research	10 (38)
Yes, with a non-academic institution (business, government, NGO ^2^, etc.)	8 (31)
Yes, with a funder	0 (0)
No	11 (42)

^1^ Respondents were asked to select all responses that apply. ^2^ NGO, non-governmental organization.

**Table 3 ijerph-18-01518-t003:** Climate action at different areas.

Area	Respondents, No.	Level 1 ^1^	Level 2	Level 3	Level 4	Level 5	Other	None
		Response, No. (%)						
Vision	27	15 (56)	2 (7)	6 (22)	2 (7)	1 (4)	1 (4)	N/A ^2^
Strategy	24	14 (58)	8 (33)	1 (4)	1 (4)	0 (0)	0 (0)	N/A
Personnel	45	31 (69)	1 (2)	6 (13)	4 (9)	1 (2)	0 (0)	2 (4)
Networks	44	26 (59)	8 (18)	3 (7)	3 (7)	1 (2)	0 (0)	3 (7)
Culture	44	13 (30)	10 (23)	4 (9)	1 (2)	3 (7)	0 (0)	13 (30)
Physical environment	45	17 (38)	11 (24)	8 (18)	2 (4)	2 (4)	0 (0)	5 (11)
Communication	45	26 (58)	8 (18)	4 (9)	3 (7)	0 (0)	0 (0)	4 (9)
Education	45	33 (73)	4 (9)	3 (7)	2 (4)	2 (4)	0 (0)	1 (2)
Research	45	38 (84)	1 (2)	1 (2)	1 (2)	2 (4)	1 (2)	1 (2)
Innovation	44	20 (45)	12 (27)	0 (0)	3 (7)	1 (2)	1 (2)	7 (16)

^1^ The climate action levels are Level 1: Ad hoc; Level 2: Cohesive; Level 3: Systematic; Level 4: Collaborative; Level 5: An example for others. See Table A1 for the corresponding statements. ^2^ N/A, non-applicable.

**Table 4 ijerph-18-01518-t004:** Measures taken by schools to reduce its ecological footprint.

Measure ^1^	Response, No. (%)
A net zero carbon building	5 (11)
Fossil fuel divestment or a fossil fuel free school	4 (9)
Green travel policy	16 (36)
A single-use plastic free building	6 (13)
Avoiding waste	25 (56)
Improving waste management	34 (76)
Increase re-use, repair or recycling	25 (56)
Using waste from some production processes as a resource in others	2 (4)
Increase biodiverse green space	7 (16)
Promoting efficient & economical water supply & use	10 (22)
Raising awareness among students (not by formal education)	28 (62)
Raising awareness among personnel	26 (58)
Providing students with the opportunity to be involved in sustainability	17 (38)
Providing personnel with the opportunity to be involved in sustainability	18 (40)
Other	1 (2)
None of these	2 (4)

^1^ Respondents were asked to select all responses that apply.

## Data Availability

The data presented in this study are available in the article.

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
