# Peer review of "Climate Action at Public Health Schools in the European Region"

_ijerph, 2021, doi:10.3390/ijerph18041518_

Round 1

Reviewer 1 Report

The paper "Climate Action at Public Health Schools in the European Region" addresses the following question: “How can the Association of Schools of Public Health in the European Region (ASPHER) best support the goals of the European Green Deal through its network of public health schools and departments?” via an online survey among ASPHER (Association of Schools of Public Health in the European Region) members. 

The study is mainly an analysis of the answers to the survey. 

If the paper achieved the aim it gives to itself and adequately presents the results of the survey, we can wonder whether it provides a deep and informed contribution to the field. It seems to us rather limited in scope.

First, the overall response rate was only 51%. So it's difficult to provide a comprehensive picture of the situation of ASPHER members engagement with Climate Change related curriculum. The fact that 64% of the 51% offer climate and health education courses could mean that only one quarter of ASPHER schools do so. We do think additional information and another survey conducted specifically on non-respondents ASPHER members should be done.

Second, the paper is conceptually speaking not clear enough. Some assumptions should be addressed and background information is missing.

  1. The articulation between Climate Change and Health Issues could be presented in a much more articulated way - to say simply we are not sure about the relations between them is not very convincing. The sentence: "In the ‘State and Outlook 2020’ report, the European Environment Agency states that the link between climate change and health is complex [3]: the gaps and uncertainties in evidence make it arguable whether negative health impacts can be reduced by current policies" could lead us to consider with skepticism the study itself. 1a. "Complex" is rather vague. Especially regarding the already well-known effects of pollution on health, the authors should be more accurate with regard to the effects of Climate Change on Health in European countries. The point being: well-known negative effects of pollution on health are already difficult to address in EU, so what can we really expect for much more complex effects of Climate change on health?   1b. These gaps and uncertainty regarding current policies should be defined more precisely. The paper should demonstrate how these shortcomings could be addressed by the new kind of curriculum the authors are advocating for. 
  2. The paper is based on the assumption of a relation between delivering courses at university level and improving Climate Change related health issues. However the articulation between the two things is not very clear. Is it because there will be better prevention of Climate Change induced diseases or because there will be better knowledge to "combat" Climate Change? Then if it's the case, the link between knowledge and practice should be better articulated. For example: "Since European Union Member States were responsible for 7.34% of global greenhouse gas emissions [4], these are steps that we encourage other schools to implement. Notable is that we found that five schools have a net-zero carbon building, which is a needed practice: buildings are responsible for 40% of the consumed energy and 36% of carbon emission in the European Union" True but what relations exactly with health issues? Why zero net building is a more pressing issue for ASPHER schools than for schools of urbanism, architecture, etc.? 

Reviewer 2 Report

You have done a thorough job laying out your argument and clearly address the issues raised by your results. You close these out well in your discussion. There are some very minor grammar issues but none that I feel need amending. This is a timely and important piece of research.

Reviewer 3 Report

You did a very interesting work on a very important topic. Some points would deserve a longer analysis in discussion and/or conclusion - What about the risk of greenwashing ? - Why the strategic vision is often missing ? Do we miss managers of sustainable development in schools of publlic health ? - Why a stand-alone course is better than a integrated component for that topic ? I don't have the answers, but I think it could be interesting to discuss it.

Round 2

Reviewer 1 Report

The reviewer's comments were not really taken into account.

As far as I can say, the revised paper did not really take into account most of the above reported comments.
Apart from the fact, that I think that the overall response rate was only 51% and that the authors could include more data after further inquiries, my concerns remains: The authors state "the gaps and uncertainties in evidence make it arguable whether negative health impacts can be reduced by current policies". But 1. These gaps and uncertainty regarding current policies are not defined precisely enough and 2. The paper should demonstrate how these shortcomings could be addressed by the new kind of curriculum the authors are advocating for.
And also this question: "Since European Union Member States were responsible for 7.34% of global greenhouse gas emissions [4], these are steps that we encourage other schools to implement. Notable is that we found that five schools have a net-zero carbon building, which is a needed practice: buildings are responsible for 40% of the consumed energy and 36% of carbon emission in the European Union" True but what relations exactly with health issues? 

Author Response

Dear reviewer,

We thank you again for your feedback on our paper "Climate Action at Public Health Schools in the European Region". Hereby, we would like to address your remaining questions.

“The authors state "the gaps and uncertainties in evidence make it arguable whether negative health impacts can be reduced by current policies". But 1. These gaps and uncertainty regarding current policies are not defined precisely enough…”

Thank you for this comment, we have tried to address your point in the discussion as follows:

  • “In a recent narrative synthesis, Lee et al. state that the accuracy of reported knowledge about the impacts of climate change varies according to the method employed and that there are erroneous ideas and misconceptions about causes of climate change [23]. This results in the fragmentation and the lack of integration of the knowledge on different aspects of a climate issues which greatly affects its usability in policymaking process [24].” (Lines: 358­–361)

    “23. Lee, K.; Gjersoe, N.; O'Neill, S.; Barnett, J. Youth perceptions of climate change: A narrative synthesis. WIREs Clim Change 2020, 11:e641, doi.org/10.1002/wcc.641.
    Williams, D.S.; Rosendo, S.; Sadasing, O.; Celliers, L. Identifying local governance capacity needs for implementing climate change adaptation in Mauritius. Climate Policy2020, 20(5), 548-562, doi: 10.1080/14693062.2020.1745743.” (Lines: 488–494)

“…and 2. The paper should demonstrate how these shortcomings could be addressed by the new kind of curriculum the authors are advocating for.”

Thank you for your feedback. To address the comment and further clarify the point of climate and health education, we have added the following:

  • “Therefore, interprofessional, interdisciplinary and transformational educational activities, which require not only problem-solving skills but also system thinking, change and implementation strategies should be included in modern curricula addressing climate change.” (Lines: 366–368)

“And also this question: "Since European Union Member States were responsible for 7.34% of global greenhouse gas emissions [4], these are steps that we encourage other schools to implement. Notable is that we found that five schools have a net-zero carbon building, which is a needed practice: buildings are responsible for 40% of the consumed energy and 36% of carbon emission in the European Union" True but what relations exactly with health issues?”

We believe our manuscript explains in the introductory part (especially in lines 69-96) the expectations on future public health professionals linking this also to desired competencies and the issue of employability. Competency in climate action is part of the expected profile of a public health professional. The discussion on reducing the environmental footprint of the health sector has been very vivid and prominent during the last decade. We believe the discussion on reducing the carbon footprint of the health sector is definitely known to the target audience and reflected in the references listed in this manuscript.

Kind regards,

Rana Orhan, on behalf of all authors
